# Extrapyramidal Side Effects with Chronic Atypical Antipsychotic Can Be Predicted by Labeling Pattern of FosB and phosphoThr^34^-DARPP-32 in Nucleus Accumbens

**DOI:** 10.3390/biomedicines11102677

**Published:** 2023-09-29

**Authors:** Sonia G. Prieto, Maria Camila Almeida, João C. S. Silva, Elaine Del-Bel, Marcela B. Echeverry

**Affiliations:** 1Center for Mathematics, Computation and Cognition, Federal University of ABC, São Bernardo do Campo 09606-045, SP, Brazil; soniaprieto@ufabc.edu.br (S.G.P.); joao.silva@aluno.ufabc.edu.br (J.C.S.S.); 2Center for Natural and Human Sciences, Federal University of ABC, São Bernardo do Campo 09606-045, SP, Brazil; camila.almeida@ufabc.edu.br; 3Department of Morphology, Physiology and Basic Pathology, Dental School of Ribeirão Preto, University of São Paulo, Ribeirão Preto 05508-000, SP, Brazil; eadelbel@usp.br; 4Neuroscience Laboratory, School of Medicine, Universidad de Santander (UDES), Bucaramanga 39006-39005, Santander, Colombia

**Keywords:** hypokinetic movement disorders, olanzapine, clozapine, haloperidol, striatum, phosphoThr^75^-Darpp-32

## Abstract

Extrapyramidal side effects (EPS) can be induced by neuroleptics that regulate the expression of transcription factor FosB and dopaminergic mediator DARPP-32 in the striatum. However, the long-term neurobiological changes in striatal projection neurons resulting from a cumulative dosage of typical and atypical antipsychotics are poorly understood. The present study aimed to determine the differential and long-lasting changes in FosB distribution and DARPP-32 phosphorylation in the striatum and nucleus accumbens (NAc) associated with chronic antipsychotic-induced EPS. Male C57Bl/6J mice received daily injections of Olanzapine (Olz, 15 mg/kg), Clozapine (Clz, 20 mg/kg), or Haloperidol (Hal, 1 mg/kg), for a period of 11 weeks with a 4-day withdrawal period before the last dosage. Catalepsy for detection of EPS, along with open-field and rotarod tests, were assessed as behavioral correlates of motor responses. Additionally, FosB and phosphorylated-DARPP-32 immunohistochemistry were examined in striatal regions after treatment. All antipsychotics produced catalepsy and reduced open-field exploration, such as impaired rota-rod performance after Olz and Hal. The washout period was critical for Clz-induced side effects reduction. Both Olz and Clz increased FosB in NAc Shell-region, and phosphoThr^34^-DARPP-32 in NAc. Only Clz reduced phosphoThr^75^-DARPP-32 in the dorsal striatum and showed FosB/phosphoThr^34^-Darpp-32-ir in the NAc Core region. This study provides evidence that atypical antipsychotics such as Olz and Clz also give rise to EPS effects frequently associated with a cumulative dosage of typical neuroleptics such as Hal. Nevertheless, FosB/phosphoThr^34^-Darpp-32-ir in the NAc Core region is associated with hypokinetic movements inhibition.

## 1. Introduction

The prevalence of extrapyramidal symptoms and side effects in patients following prolonged use of typical or first-generation antipsychotics (FGAs) led to the introduction of atypical or second-generation antipsychotics (SGAs). These are preferentially prescribed to patients susceptible to developing extrapyramidal side effects (EPS), such as dystonia, parkinsonism, akathisia, and tardive dyskinesia (TD), as well as to those not responding adequately to other treatments [1,2,3,4]. EPS are a leading cause of noncompliance with antipsychotic medication which is associated with a relapse of psychotic symptoms and re-hospitalization [5]. In clinical practice, treating schizophrenia remains a significant challenge, especially in patients who do not respond to or cannot tolerate standard antipsychotic therapy [6]. Physicians have limited treatment strategies in such cases [6].

The potential causes of EPS symptoms, as measured by catalepsy test in animal models, have been attributed to D2 receptor supersensitivity, striatal neurodegeneration, maladaptive synaptic plasticity, enhanced serotonin 5-HT_2_ receptor signaling, and a transient central D2 occupancy [7,8,9,10,11,12,13,14].

Clozapine (Clz), the prototypical atypical antipsychotic, referred to as second-generation antipsychotics, binds 10 times stronger to the D4 dopamine (DA) receptor subtype than to D2 and D3 receptors [15], shows high affinity for 5-HT_2_ sites and displays inverse agonist activity in the 5-HT_2A_ receptor [16], with a much lower affinity for D2 sites in the cerebral cortex and striatum [17]. Besides, its agonistic activity to 5-HT_1a_ receptors is linked to its effectiveness in treating negative symptoms of schizophrenia with minimal EPS liability and absence of cataleptogenic properties [18,19]. Olanzapine (Olz) is another atypical antipsychotic with a pharmacological profile resembling Clz [20]. It shows a greater affinity for several serotoninergic receptors than for the D2 receptors [21]. Both Olz and Clz elevate cortical DA levels and norepinephrine outflow to a greater extent than Hal, suggesting potential for greater clinical improvement in cognitive function independently of mesolimbic dopaminergic terminals [22]. 

Haloperidol (Hal), the classic typical antipsychotic, referred to as first-generation antipsychotics, is well known for its high risk of inducing EPS [23]. It is characterized by its antagonistic binding to dopamine D2 receptors (D2Rs)-expressing striatal medium spiny neurons (MSNs), which form the striatopallidal pathway, also known as the indirect pathway [24]. 

Different induction patterns of Fos-like immunoreactivity have been used in antipsychotic-induced EPS: it is suggested that an increase in Fos, mainly in the dorsolateral striatum (DL-striatum) may be a predictor of EPS [25,26]. Indeed, second-generation antipsychotics failed to increase Fos-like in this region [25]. 

Both categories of antipsychotic drugs exhibit varying degrees of antagonistic affinity for dopaminergic D2 receptors. In MSNs of the indirect pathway, the D2 antagonistic activity of antipsychotic drugs is maintained through the action of Adenosine Receptor A2A (A2ARs) coupled with G_olf_-mediated activation of adenylyl cyclase and protein kinase A (PKA) [27,28]. PKA increases phosphorylation of the cAMP response element-binding protein (CREB) to control c-Fos and FosB/ΔfosB (Fos-like) expression [29]. Furthermore, PKA can increase the phosphorylation of the GluR1 subunit in the glutamate AMPA receptor [30], as well as the histone H3 and 32 kDa dopamine- and cyclic AMP-regulated phosphoprotein (DARPP-32) [31].

Previous investigators have demonstrated that the postsynaptic effects of DA are conferred to DARPP-32 in striatal projection neurons [32]. Activation of PKA or protein kinase G (PKG) stimulates DARPP-32 phosphorylation at Thr^34^ and thereby converts DARPP-32 into a potent inhibitor of Protein Phosphatase-1 (PP-1) [31]. DARPP-32 is also phosphorylated at Thr^75^ sites by Cdk5, which subsequently converts DARPP-32 into an inhibitor of PKA [33]. Besides, the DARPP-32 protein regulates the phosphorylation levels and thereby the activity of a broad range of target proteins [34]. In addition to DA, numerous neurotransmitters, peptides, and neuromodulators have been shown to provide input to DARPP-32-associated signal transduction cascades [34]. These results highlight the importance of DARPP-32 as a critical component for signal integration in MSNs [35]. Bateup and co-workers [36] showed that the deletion of DARPP-32 in indirect MSNs produces a robust increase in locomotor activity and a decrease in Hal-induced catalepsy, whereas DARPP-32 inactivation in direct MSNs can abolish dyskinetic behaviors in response to the Parkinson’s disease drug L-DOPA [36].

Dopamine D2R blockade is preserved in all currently available antipsychotics and is essential for their effectiveness. While there is a certain decrease in the risk of EPS with second-generation antipsychotics, the susceptibility to EPS must be weighed against the acknowledged potential for metabolic syndrome [37,38,39]. Besides, in medical practice, a considerable number of individuals with schizophrenia do not respond adequately to antipsychotic medication alone, often requiring a combination of second-generation antipsychotics [40,41]. In this situation, the addition of a second antipsychotic has shown different outcomes needing more rigorous randomized trials [40].

Then, the importance of separately studying downstream effector proteins in MSNs becomes evident due to variations in the occurrence of EPS among these types of antipsychotics, which may play a role in unfavorable synergistic actions during dual therapy. These three medications show different affinity and dissociation from D2R, and by comparing their effects, we aim to gain insights into potential variations in treatment response and side effects among individuals. This information is critical for optimizing treatment strategies. Comparing the differences in measurements of extrapyramidal symptoms between Olz versus Clz is also crucial, and Hal is associated with safety liabilities. 

One such line of research, yet not well-understood, is the analysis of the FosB/phosphorylated Thr^34^ or Thr^75^-DARPP-32 signaling pathway in reduced motor movements, or EPS following chronic treatment with Olz, Clz, or Hal. The goal of the present investigation is, therefore, to determine what differential and long-lasting changes in FosB and DARPP-32 phosphorylation immunoreactivity (-ir) are present in the striatum and Nucleus accumbens (NAc) following hypokinetic movement disorders associated with antipsychotic-induced EPS.

## 2. Materials and Methods

### 2.1. Animals

The thirty-two-adult male C57Bl/6J mice, were 8 weeks old at the beginning of the experiments, weighing 25–30 g. Animals were obtained from Jackson Lab’s colonies (Sacramento, CA, USA) housed at UFABC Animal House in a temperature-controlled room (23 °C) with a 12 h light/dark cycle (lights on at 7:00 a.m.) and with unlimited access to water and food. Animals were weighted on a weekly basis throughout the study and no difference in weight gain was observed among animals that received different treatments. 

Handling of all animals and experimental protocols were in accordance with the guidelines outlined by the Committee for Animal Use of UFABC (CEUA-UFABC) and approved under protocol #014/2013. All efforts were made to minimize animal suffering and the number of animals used. Experiments were performed between 8:00 and 12:00 a.m. in a soundproofed, temperature-controlled room, lit with two 40 W fluorescent lights placed 1.3 m away from the animal. The experimental apparatuses were cleaned with 10% ethanol solution and dried before each session.

### 2.2. Drug Administration

Olanzapine (Olz, 15 mg/Kg, Zyprexa tablet^®^-Elly Lilly company, SP, Brazil, diluted in 25% DMSO—saline, *N* = 8) [42,43], Clozapine (Clz, 20 mg/Kg, Leponex tablet^®^-Novartis, São Paulo, SP, Brazil, diluted in 25% DMSO—saline, *N* = 8) [3,43,44], Haloperidol (Hal, 1 mg/Kg, Janssen Drops^®^-Janssen–Cilag Farmaceutica, São Paulo, SP, Brazil), diluted in saline, *N* = 8) [43,44,45]; 25% DMSO solution (DMSO solution, Zoetis^®^, São Paulo, SP, Brazil, for veterinary use in a saline solution, *N* = 4) or saline solution (Sal, vehicle of Hal, *N* = 4) were injected intraperitoneally (i.p.) in a volume of 10 mL/kg. This concentration of DMSO (25%) resulted in no adverse reactions and a lack of motor side effects. The doses used for Clz and Hal fall within a human dosage range (see calculation in [46]). For Olz, the dose was based on toxicokinetic studies in rats. Chronic treatment was carried out via continuous daily injections for a period of 10 weeks (days 1–72) followed by a 4-day withdrawal period [12] from days 73–76. At day 77, all groups received a last dose (identical to the treatment before withdrawal) twenty-four hours before perfusion (day 78, Figure 1A), in such a way that stable EPS could be evaluated in response to the last injection after the withdrawal period. It has been reported that TD rarely emerges during continued antipsychotic treatment, but severe complications can occur after an abrupt withdrawal period [47,48,49].

### 2.3. Behavioral Test

Catalepsy, open-field, and rota-rod tests were carried out 90 min after injection, on a weekly basis for three consecutive days, with a 24-h interval between each test. The catalepsy test was conducted starting from the first week of treatment onwards. Locomotor activity and the rota-rod test were performed beginning in the fifth week of treatment (Figure 1A), following the observation of a lack of tolerance development to the cataleptic effect through repetitive administration of antipsychotics. All treatment groups were tested under the same trial conditions. The proposed doses used in this study have been previously reported to induce vacuous chewing movements (VCMs) [42], aligning with one of the initial objectives of this study. While this behavior was eventually observed, its consistency was lacking, thereby precluding a comprehensive analysis.

#### 2.3.1. Catalepsy Test

Each animal was injected with the test compound or vehicle (i.p.) and the catalepsy procedure was evaluated using the bar test. This test consists of placing both of the animal’s forelimbs over a horizontal cylindrical glass bar (diameter: 0.5 cm; height: 4.5 cm above the table) [50]. The time (in seconds—s) in which both forelimbs remained on the bar was recorded up to 300 s [51]. Catalepsy was defined as having terminated when at least one forepaw touched the floor or when the mouse climbed onto the bar.

#### 2.3.2. Open-Field Spontaneous Locomotor Activity

General exploratory activity was evaluated in an open field test as described by Del Bel and co-workers [52]. The experimental apparatus consists of a square cage measuring 46 × 46 cm, with 30 cm high walls, featuring an open ceiling. The EthoVision XT 11.5—Noldus software platform (Leesburg, VA, USA) was used to score locomotion (defined as the number of floor areas entered with all four paws). The mice were placed in the center of the open field [52] and their behavior was recorded up to 300 s. Immobility time and the number of crosses made across the boundary lines of the floor zones were recorded and further analyzed. Immobility time was defined as the duration a mouse remains in a zone without performing grooming, rearing, or exploring the environment. Animals were transported from the housing area to the testing room 30 min before testing.

#### 2.3.3. Rota-Rod Test

Motor coordination and balance were assessed using a rota-rod test (Panlab^®^ model LE8500, Barcelona, Spain). The rota-rod test involved placing the mouse on a rotating drum and measuring the time it took for each animal to maintain its balance while walking on top of the rod. Animals were pre-trained twice a day for 3 days before the test. The speed of the rota-rod accelerated from 4 to 10 revolutions per minute (rpm) over a 300 s period. To ensure alertness during each session, the experimenter gently tapped the animals’ tails at times [53]. Rota-rod tests were conducted on separate days from the catalepsy and open field tests.

### 2.4. Tissue Preparation

Studies involving chronic administration of dopaminergic drugs have shown that twenty-four hours after the last dose, the levels of FosB/ΔfosB immunoreactivity remain elevated due to longer-lasting transcriptional changes in both NAc and the dorsal striatum [54]. Therefore, this specific time point was established in our study. On day 78, the mice were anesthetized with 30% urethane (Sigma-Aldrich, St. Louis, MO, USA) and transcardially perfused with 200 mL of 0.01 M PBS followed by 200 mL of 4% paraformaldehyde (Sigma-Aldrich, St. Louis, MO, USA) in a 0.1 M phosphate buffer (pH 7.4). The brains were rapidly removed and post-fixed in 4% paraformaldehyde for 2 h (h) and then cryoprotected in a 30% sucrose/phosphate buffer for 48 h (4 °C). Brains were quickly frozen in cold isopentane (−40 °C; Sigma-Aldrich, St. Louis, MO, USA) and stored at −80 °C for further analyses. The brains were sectioned into 30 µm slices using a cryostat (CM1850 Leica Microsystems^®^, Wetzlar, Germany). Sections taken through the striatum were collected in ethylene glycol anti-freezing solution and stored at −20 °C until the day of the immunohistochemistry assay [43].

#### Immunohistochemistry

FosB and DARPP-32 immunostaining were performed on different and adjacent brain sections containing Caudate—Putamen (Cpu or striatum) and NAc, along an anterior–posterior (AP) axis. Immunohistochemistry was performed using a standard free-floating peroxidase-based method [43]. To reveal the antigens and epitopes, a citrate buffer antigen retrieval protocol was used. The sections were incubated with rabbit polyclonal antibody to FosB (1:1000 sc-7203 (H-75), Santa Cruz Biotechnology, Santa Cruz, CA, USA), or rabbit polyclonal antibody to DARPP-32 (phosphoThr^34^ or phosphoThr^75^, 1:400 Abcam^®^, Cambridge, UK ab51076 and ab51114, respectively) [35,55] The FosB antibody detects both FosB and ΔFosB which are particularly significant for long-term modifications in the nervous system [56,57,58]. To accurately quantify the increase in DARPP-32 positive cells in the present study, it was essential to establish a positive control in all striatal sections followed by analysis of the motor cortex immediately above the striatum. Three assays were performed to stain mouse DARPP32, and they did not differ in their immunohistochemical staining for phosphoThr^34^ or Thr^75^, showing specific labeling in cortex, and striatum (Figure 4C).

Incubation with these antibodies was performed overnight at 4 °C followed by incubation with biotinylated secondary goat anti-rabbit IgG antibody and peroxidase (Vectastain^®^ Elite^®^ ABC-HRP Kit, PK-6100, Vector Laboratories, Southfield, MI, USA). The sections were developed using 3,3’-diaminobenzidine (DAB; Sigma-Aldrich, St. Louis, MO, USA, 1 mg/mL) and hydrogen peroxide (0.03%). For DARPP-32, 0.5% nickel was added to visualize the cells with greater contrast. Tissues from all experiments were always processed in the same assay and a negative control sample without the primary antibody was included. The slices were mounted onto slides and cover-slipped for microscopic observations.

### 2.5. Quantification of Immunohistochemistry

A preliminary qualitative analysis of all brain sections was conducted to identify bilateral labeling and brain structures. Structure localization was determined with the help of the Paxinos and Franklin atlas [59]. Images containing immunostained nuclei or cells from slices were digitally captured with 10 X magnification through a Leica DFC295 camera connected to the 5500 DM B Leica^®^ microscope. A computerized image analysis system (ImageJ, NIH System—https://imagej.nih.gov/ij/download.html, for Windows, 64-bit Java 8, January 2019) was used. The striatum was subdivided into three regions for analysis: dorsolateral (DL), dorsomedial (DM), and ventrolateral (VL) (Figure 2F) [60]. The NAc was subdivided into core and shell regions (Figure 2F) [61]. For FosB/ΔfosB, immunoreactivity (ir) labeling was visualized by a higher brown reaction product, and cytoplasmic DARPP-32 positive cells were visualized as a dark blue reaction product [35]. At least three bilateral sections from each animal were selected for counting the number in each region of positive nuclei or cells, according to corresponding coordinates on the atlas along an anterior–posterior (AP) axis (bregma 0.86, 0.74, and 0.62 mm). The counting was automated using ImageJ following the basic image processing from the software. To ensure the validity of the results and minimize bias, two different experimenters conducted the counting independently while blinded to the data. The results were subsequently cross-validated for accuracy. Results are expressed as the mean density of cells (number of positive nucleus or cells/0.1 mm^2^ of the region) calculated from results obtained in each brain side [62]. Four animals per group were included based on established criteria for IHS and quantification. The quantitative analysis was conducted blindly.

### 2.6. Statistical Analyses

The data were tested for unequal variance and normality, requiring a log-transformation (with the addition of a constant value of 1 for catalepsy results) [43,51]. Measures of EPS induced by chronic treatment, as assessed by the three behavioral tests (catalepsy, open field, and rota-rod), were analyzed with two-way repeated-measures ANOVA with Weeks (11 levels) as a within factor and Treatment as a between-subjects factor. A one-way ANOVA was conducted to assess differences in the number of positive nuclei and cells within specific striatal and NAc regions for the atypical groups and values of estimates of effect size with ω^2^ between 0.5 and 0.8 were confirmed. T-tests were employed to compare the mean number of cells labeling for the Hal and Sal conditions. The Bonferroni test was used for multiple comparison procedures between the treatment conditions and their respective control group. A significance level of *p* < 0.05 was considered statistically significant. All values are expressed as mean ± standard error of the mean (S.E.M.). The Pearson correlation coefficient (Pearson’s r, range of values from +1 to −1) was used to study the association between each behavioral test and cell immunoreactivity in striatal and NAc regions with significant changes. Linear regression was employed after calculating the correlation coefficient to evaluate the significance of each predictor. Measures of behavioral tests on day 77 (after withdrawal) were used for correlations. A significance level of *p* < 0.05 was considered. A correlation heatmap (Pearson’s r correlation matrix), based on pairwise correlations, was created using JASP. The OriginPro 8.5 program was used for scientific graphing, and JASP Version 0.16.2 was used for statistical analysis.

## 3. Results

### 3.1. Antipsychotic Treatment-Induced Extrapyramidal Side Effects (EPS)

#### 3.1.1. Catalepsy Test

As shown in Figure 1B, the atypical antipsychotic experiment exhibited a significant main effect of weeks [*F*(10,170) = 2.754; *p* = 0.004], treatment [*F*(2,17) = 14.25; *p* < 0.001] (Figure 1B(A.1)), and an interaction between weeks vs treatment [*F*(20,170) = 4.195; *p* < 0.001]. Analyzing simple main effects with week as a moderator factor revealed a treatment effect from weeks 2 to 8 (all *p* values < 0.01), and at week 11 (*p* < 0.004). Although an increase in catalepsy time was observed over time, no significant difference was found between Olz and Clz groups (Bonferroni test, *p* > 0.05); however, the Clz group exhibited increased cataleptic time from the second week (ANOVA, *F*(2,17) = 8.015; *p* = 0.004), with the cataleptic effect disappearing in the last week after withdrawal (Clz vs. DMSO, Bonferroni test, *p* > 0.05, Figure 1B(A.1)).

Additionally, in Figure 1B, the Hal experiment demonstrated a significant main effect of treatment [*F*(1,10) = 1070.76; *p* < 0.001, Figure 1B(A.2)]. There were no significant effects for weeks [*F*(10,100) = 1.533; *p* = 0.139] or interaction [*F*(10,100) = 1.553; *p* = 0.132]. Analyzing simple main effects with weeks as a moderator factor revealed a treatment effect from weeks 1 to 11 (all *p* values < 0.001).

#### 3.1.2. Open-Field Exploration

Regarding the Olz and Clz, a significant week*treatment interaction was observed [*F*(12,102) = 3.534; *p* < 0.001], along with the effect of the weeks factor [*F*(6,102) = 2.966; *p* = 0.01] and treatment factor [*F*(2,17) = 12.145; *p* < 0.001, Figure 1B(B.1)]. There was a significant decrease in total line crossings observed each week (simple main effects: effect of treatment at 5th–7th and 9th–11th weeks, all *p*-values < 0.048; Bonferroni test, *p* < 0.05), except in the 5th week for both treatments and in the 7th for Olz (Bonferroni test, *p* > 0.05). Following withdrawal, a decreased effect was only observed for the Olz group [ANOVA, *F*(2,17) = 3.659; *p* = 0.048, Bonferroni test, *p* < 0.05, Olz vs. DMSO]. A significant decrease in total line crossing was noted in the open field test with Hal treatment [week*treatment interaction: *F*(6,60) = 18,591; *p* < 0.00] along with effects of the weeks factor [*F*(6,60) = 16.960; *p* < 0.001] and the treatment factor, *F*(1,10) = 70.138; *p* < 0.001, Figure 1B(B.2)] for all weeks studied.

Immobility time was also evaluated in the open-field test. For atypical antipsychotics, interaction weeks and treatment yielded a significant result [weeks*treatment: *F*(12,102) = 2.173; *p* = 0.018, the effect of treatment, *F*(2,17) = 7.399; *p* = 0.005, Figure 1B(B.3)], with the Clz group exhibiting higher immobility with significant differences in the 7th–9th weeks (simple main effects: all *p* values < 0.034, Bonferroni test, *p* < 0.05). On the contrary, no increase in immobility time was observed for the Olz treatment (Bonferroni test, *p* > 0.05, Figure 1B(B.3)). After the withdrawal of treatment, no change was detected [ANOVA, *F*(2,17) = 2.065; *p* = 0.157]. Rearing frequency (number of times the animal stood on its hind legs) and feces number count were also measured, yet no significant differences were observed for these measures (RM ANOVA, *p* > 0.05). Animals treated with Hal displayed a marked immobility [weeks*treatment: *F*(6,60) = 2.391; *p* = 0.039, effect of Treatment, *F*(1,10) = 404.86; *p* < 0.001, Figure 1B(B.4)].

#### 3.1.3. Rota-Rod Test

As shown in Figure 1B(C.1), the use of atypical antipsychotics also results in a significant decrease in performance on the rota-rod test [weeks*treatment: *F*(12,102) = 2.257; *p* = 0.014, effect of factor weeks, *F*(6,102) = 5.460; *p* < 0.001, and effect of treatment, *F*(2, 17) = 4.255; *p* = 0.032, Figure 1B(C.1)]. An analysis of simple main effects with weeks as the moderator factor revealed an effect of treatment in the fifth (*p* = 0.039), and ninth (*p* = 0.027) weeks, with the Olz group differing from the vehicle (Bonferroni test, *p* = 0.045), but no difference was observed with the Clz group (Bonferroni test, *p* > 0.05). Following withdrawal, no significant changes were observed [ANOVA, *F*(2,17) = 2.207; *p* = 0.141]. Treatment with Hal decreased the latency to fall from the rota-rod apparatus compared to the control group [Figure 1B(C.2), weeks*treatment: *F*(6,60) = 4.116; *p* = 0.002, the effect of weeks, *F*(6,60) = 3.990; *p* = 0.002, and effect of treatment, *F*(1,10) = 49.9; *p* < 0.001].

### 3.2. Effects of Chronic Drug Treatment on FosB and DARPP-32 Immunoreactivity in Neostriatum

#### 3.2.1. FosB Labeling in the Neostriatum and NAc

Figure 2A–D depict the number of FosB ir-positive cells following antipsychotic treatments. With atypical antipsychotic treatment, a significant increase in FosB ir- positive cells was observed under the Olz treatment condition (Figure 2A,B) in the striatal DL region [*F*(2,9) = 7.96; *p* = 0.01, Bonferroni test, *p* < 0.05] and DM [*F*(2,9) = 25.866; *p* < 0.001, Bonferroni test, *p* < 0.05] compared to other groups. Within the NAc, elevated FosB labeling was observed in the Shell region for both atypical antipsychotics [*F*(2,9) = 13.791; *p* = 0.002, Bonferroni test, *p* < 0.05, Figure 2A,B], whereas in the NAc Core region an increase in FosB-ir was observed only following Clz treatment [*F*(2,9) = 7.656; *p* = 0.011, Bonferroni test, *p* < 0.05] in comparison to its respective control. No significant changes were detected the in VL region [*F*(2,9) = 4.459; *p* = 0.065]. Compared to its respective control, Hal induced an elevation in the number of FosB-ir cells in all striatal regions (DL: t = −19.121; df = 6, *p* < 0.001; DM: t = −17.075; df = 6, *p* < 0.001; VL: t = −9.411; df = 6, *p* < 0.001) and NAc (Shell: t = 11.84; df = 4, *p* < 0.001; Core: t = 34.72; df = 4, *p* < 0.001) (Figure 2C,D).

#### 3.2.2. DARPP32-Thr^34^ and DARPP32-Thr75 Labeling in the Neostriatum and NAc

The results from immunostaining experiments for phosphoThr^34^-DARPP-32 (Figure 3) or phosphoThr^75^-DARPP-32 (Figure 4) are shown for the neostriatum and NAc regions following antipsychotic treatments. In the dorsal neostriatum, an increased phosphoThr^34^-DARPP-32-ir signal was observed after Olz treatment in comparison to other experimental groups [DL: *F*(2,9) = 7.515; *p* = 0.012; Bonferroni test, *p* < 0.05; DM: *F*(2,9) = 6.223; *p* = 0.02; Bonferroni test, *p* < 0.05]. Both Olz and Clz treatments resulted in an increase in the phosphoThr^34^-DARPP-32 labeling in the Shell and Core regions of NAc [shell *F*(2,9) = 12.506; *p* = 0.003; Bonferroni test, *p* < 0.05, and core: *F*(2,9) = 8.290; *p* = 0.009; Bonferroni test, *p* < 0.05] (see Figure 3A,B). Conversely, a significant decrease in the number of neurons positive for Thr^34^-DARPP-32 was observed in the DL neostriatum following Hal treatment (t = 5.793; df = 6, *p* = 0.001), as well as in the NAc (Shell: t = 6.665; df = 6, *p* < 0.001; Core: t = 3.465; df = 6, *p* = 0.013) (Figure 3C,D). No significant changes were observed in the VL region [atypical antipsychotics, *F*(2,9) = 1.90; *p* = 0.205; Sal-Hal, t = −0.221; df = 6, *p* = 0.832].

For Thr^75^-DARPP-32, statistical analysis indicated decreased staining after Clz in the DL [*F*(2,9) = 10.81; *p* = 0.004; Bonferroni test, *p* < 0.05] and DM [*F*(2,9) = 7.844; *p* = 0.011; Bonferroni test, *p* < 0.05, Figure 4A,B] compared to other experimental groups. No significant differences were observed in the immunohistochemical analysis of Thr^75^-DARPP-32 in the striatum following Hal treatment (Figure 4A, DL: t = 1.867; df = 6, *p* = 0.11; DM: t = −0.210; df = 6, *p* = 0.84; Core: t = 0.007; df = 6, *p* = 0.99; Shell: t = 0.784; df = 6, *p* = 0.46), and no significant changes were detected in the remaining striatal regions following antipsychotic treatment.

Although each antipsychotic group was compared with its respective vehicle, an increased basal level of FosB was observed in the dorsal striatum and NAc for the DMSO group. However, the observed increase after the administration of atypical antipsychotics was nearly or above fifty percent (Figure 2A,B). Similarly, to FosB labeling, basal levels of phosphoThr^34^- and Thr^75^-DARPP32 were also detected in the vehicle groups (Sal and 25% DMSO, Figure 3 and Figure 4) in the dorsal striatum and NAc. Nonetheless, the significant changes observed in labeled neuronal cells ranged from forty to ninety percent when compared to their respective vehicles.

To establish a positive control for the quantification of DARPP-32+ cells, we selected a brain region presumably to be dopaminergic. For this purpose, sections from the motor cortex, located immediately above the striatum were analyzed. The qualitative analysis confirmed the presence of DARPP-32+ cells in all sections [34]. In Figure 4C, photomicrographs confirming specific labeling of phosphoThr^34^-DARPP-32 and phosphoThr^75^-DARPP-32 in the Cortex (Cx) and Striatum (Str) are displayed [35].

### 3.3. Correlations

Schematic representations providing an overview of behaviors (Figure 5A), and significant alterations in protein-ir compared to controls (Figure 5B) are also presented. Subsequently, we conducted Pearson’s r analysis for subsets of each behavior and each protein-ir (heatmap, with correlation coefficient (r), Figure 5C). Furthermore, a significant simple linear regression analysis (ANOVA) was performed as described below.

Although Olz did not exhibit a significant immobilization effect (Figure 5A–C, Olanzapine), a noteworthy positive correlation was observed between immobilization time and FosB in the DL striatum [F(1,6) = 7.42, *p* = 0.034], NAc Shell [F(1,6) = 42.61, *p* < 0.001], and phosphoThr^34^-DARPP-32, in the NAc Shell [F(1,6) = 6.910, *p* = 0.039]. Only Olz consistently reduced total line crossing after withdrawal, displaying a significant negative correlation with FosB in the striatum [DL, F(1,6) = 13.43, *p* = 0.011; DM, F(1,6) = 7.34, *p* = 0.035], NAc Shell [F(1,6) = 17.19, *p* = 0.006], and significant negative correlation with phosphoThr^34^-DARPP-32 in the NAc [Shell, F(1,6) = 11.56, *p* = 0.014; Core, F(1,6) = 7.86, *p* = 0.031]. Olz-induced catalepsy was confirmed after withdrawal and displayed a significant positive correlation with FosB in the NAc Shell [shell, F(1,6) = 11.35, *p* = 0.015]. While Olz induced a significant decrease in rota-rod activity, with this effect disappearing after withdrawal, no significant correlation with any protein labeling was observed. Most relevant, immunoreactivity and simple linear regression analysis revealed that the absence of an immobilization effect induced by Olz significantly correlates with an increase in FosB-positive cells in the DL striatum and NAc Shell, as well as an increase in phosphoThr^34^-DARPP-32 in the NAc-Shell. Catalepsy induced by Olz also correlates with an increase in FosB-ir in the NAc-Shell region. Furthermore, for Olz, the decrease in total line crossing correlated with an increase in FosB-ir in the dorsal striatum and NAc-Shell, as well as with an increase in phosphoThr^34^-DARPP-32 in the NAc (Figure 5A–C, Olanzapine).

For Clz treatment (Figure 5A–C, Clozapine), a significant positive correlation was detected between immobilization time with FosB in the NAc [Shell, F(1,6) = 17.18, *p* = 0.006; Core, F(1,6) = 10.49, *p* = 0.018]; with phosphoThr^34^-DARPP-32, in the NAc Shell [F(1,6) = 13.14, *p* = 0.011]; and a significant negative correlation with phosphoThr^75^-DARPP-32 in the DL [F(1,6) = 13.36, *p* = 0.011]. Although Clz induced an immobilization effect, this effect was not observed on the last day after withdrawal. This loss significantly correlated with the increase in FosB-positive cells in the NAc, the increase in phosphoThr^34^-DARPP-32 in the NAc Shell, and the decrease in phosphoThr^75^-DARPP-32 in the DL region (Figure 5A–C, Clozapine). Nevertheless, a correlation with the long-term increase before the withdrawal cannot be ruled out. Additionally, Clz induced a catalepsy behavior and a decrease in the total lines crossed in an open field, with the loss of these effects after withdrawal (Figure 5A), although no significant correlations were detected with any protein and this result.

In contrast to atypical antipsychotics, Hal (Figure 5A–C, Haloperidol), displayed a significant positive correlation between catalepsy behavior and FosB in the entire striatum [DL, F(1,6) = 443.1, *p* < 0.001; DM, F(1,6) = 170.4, *p* < 0.001; VL, F(1,6) = 119.5, *p* < 0.001], as well as in the NAc [Shell, F(1,6) = 65.62, *p* < 0.001; Core, F(1,6) = 106.1, *p* < 0.001]. A similar pattern was observed for Immobilization time in the striatum [DL, F(1,6) = 23.26, *p* = 0.003; DM, F(1,6) = 27.53, *p* = 0.002; VL, F(1,6) = 13.07, *p* = 0.011], and NAc [Shell, F(1,6) = 20.32, *p* = 0.004; Core, F(1,6) = 32.6, *p* = 0.001]. Regarding phosphoThr^34^-DARPP-32, both behaviors showed a significant negative correlation in the DL [catalepsy: F(1,6) = 45.75, *p* < 0.001; immobilization time: F(1,6) = 10.64, *p* = 0.017], as well as in the NAc [catalepsy: Shell, F(1,6) = 36.58, *p* < 0.001; Core, F(1,6) = 15.97, *p* = 0.007; immobilization time: Shell, F(1,6) = 11.79, *p* = 0.014].

Furthermore, with Hal, in contrast to catalepsy and immobilization, a significant negative correlation was detected with FosB in the whole striatum and NAc for total line crossing [DL, F(1,6) = 17.67, *p* = 0.006; DM, F(1,6) = 17.65, *p* = 0.006; VL, F(1,6) = 13.83, *p* = 0.01; and NAc: Shell, F(1,6) = 17.24, *p* = 0.006; Core, F(1,6) = 20.15, *p* = 0.004], as well as rota-rod activity [DL, F(1,6) = 17.67, *p* = 0.006; DM, F(1,6) = 17.65, *p* = 0.006; VL, F(1,6) = 13.83, *p* = 0.01; and NAc: Shell, F(1,6) = 17.24, *p* = 0.006; Core, F(1,6) = 20.15, *p* = 0.004]. For both Total lines crossing and rota-rod activity, positive correlations were found for phosphoThr^34^-DARPP-32 [Total lines crossing in DL: F(1,6) = 6.87, *p* = 0.04, NAc, Shell: F(1,6) = 9.69, *p* = 0.021, NAc Core: F(1,6) = 6.16, *p* = 0.048, and rota-rod performance in DL: F(1,6) = 7.31, *p* = 0.035, and NAc-Shell: F(1,6) = 8.166, *p* = 0.029].

In summary, with Hal (Figure 5A–C), the whole striatum and NAc appear to participate in all behaviors, significantly correlating with the increase in FosB in these regions, but in different directions: positively for catalepsy and immobilization time, and negatively to Total lines crossing and rota-rod performance. An opposing pattern was observed for phosphoThr^34^-DARPP-32: a reduction of the protein-ir in the DL and NAc-Shell regions negatively correlates with catalepsy and immobilization time, while it positively correlates with Total lines crossing, and rota-rod performance.

## 4. Discussion

It is reported that typical and atypical antipsychotics differ in their ability to induce EPS. Antipsychotic-related extrapyramidal symptoms result from the disruption of dopamine neurotransmission in the brain, particularly the blockade of D2 dopamine receptors. These symptoms can vary in type and severity and may occur shortly after starting treatment or, in the case of tardive dyskinesia, after prolonged use of antipsychotic medications.

In the present study, EPS were observed after atypical antipsychotics [42,44], with no tolerance development in our experimental conditions. In this study, we reported sustained EPS and hypokinetic behaviors following chronic Olz and Clz antipsychotic administration. Nevertheless, Olz treatment did not result in immobility, as assessed in the open field test, and Clz treatment showed no significant changes in rota-rod performance.

The absence of a cataleptic effect in the last week after withdrawal was detected only with Clz treatment, while Olz and Hal continued to show this effect. Furthermore, after withdrawal, Clz revealed a loss of effect on the behaviors assessed in the open-field test (total line crossing and immobilization time), while Olz showed a loss of effect in the rota-rod activity. Conversely, no behavioral changes were evidenced with Hal treatment after the withdrawal period. Despite the presence of hypokinetic behaviors observed with atypical antipsychotics, the tendency was for them to disappear after the washout period, whereas with Hal they remained unchanged.

Although EPS and TD are highly associated with the typical antipsychotic like Hal [23], clinical trials and meta-analyses have shown that second-generation antipsychotics have the potential to induce a certain degree of EPS, such as akathisia [63,64] requiring regular monitoring of patients for both tolerability and effectiveness, as compared to first generation antipsychotics [63].

The inability of second-generation antipsychotics to produce EPS is due to their higher antagonism of 5-HT_2_ receptors compared to D2R within the mesolimbic dopaminergic pathway, involving NAc, resulting in a reduced risk for EPS [65]. Despite being designed with lower D2R affinity, after chronic treatment, this occupancy is not negligible. In fact, Olz [14] or Hal [13] result in basal ganglia D2Rs occupancy exceeding 70%, with Hal acting in the firing rate of neurons in the substantia nigra reticulate [66]. Conversely, Clz exhibits more modest central D2R occupancy—lower than 63% [67], quickly declining shortly after administration [14]. Clz’s limited EPS liability is associated with its capacity as a 5-HT_1A_ receptor agonist [18,19,68], controlling dopaminergic nigrostriatal pathway [69], and potentially counteracting Fos-positive cells in the DL striatal caused by first-generation antipsychotics, as Hal [70].

In our study, catalepsy induced by both Olz and Hal persisted even after a short withdrawal period. This suggests that rebinding might maintain higher concentrations of antipsychotics at the synaptic cleft, potentially leading to postsynaptic D2Rs occupancy. This close connection between rebinding rates and EPS could contribute to “on-target” side effects [71]. In contrast, a 4-day withdrawal period effectively reduced Clz-induced hypokinetic effects such as catalepsy and reduced line crossing and immobilization in the open field. This indicates that for Clz, the washout period might be crucial in reducing its side effects. Supporting this, FosB-ir in the dorsal striatum significantly increased after Olz and Hal, but not Clz. This could account for the reduced side effects observed after Clz withdrawal. As described later, this lack of side effects might explain the absence of a significant correlation between catalepsy or Total lines crossing with any protein.

Then, to better understand the sustained hypokinetic behavior resulting from cumulative doses of antipsychotics, we focused on the correlations of the behaviors with the two important mediators of dopaminergic signaling in MSNs: FosB and DARPP32 [28] in the different areas of the striatum and NAc. In effect, these two markers have been investigated for insights into neural networks involved in EPS, serving as a pivotal step toward designing improved therapeutic strategies [28]. Although Pearson correlations were conducted using behavioral measurements after withdrawal, the possibility of an accumulative effect over chronic treatment cannot be excluded. This is particularly relevant considering that a progressive increase in the ΔFosB and its stable product FosB have been identified as potential cellular mechanisms underlying responses to chronic drug administration with striatal effects [72].

In our linear regressions, as illustrated by the correlation matrix for each drug, a positive correlation emerges between immobilization time and catalepsy, particularly evident in the matrices for Olz and Hal, and is predicted by the increase in FosB-ir in the NAc-Shell region. In fact, this region consistently exhibited FosB-ir and a positive correlation with immobilization for all treatments. Therefore, the NAc-Shell region could potentially be linked to the persistent immobilization effect observed with Hal, in contrast to the absence of this effect with Olz, and the reduced immobility seen with Clz after withdrawal.

Furthermore, in relation to catalepsy, both the Olz and Hal exhibited a positive correlation of FosB-ir in NAc-Shell region, suggesting that this region might also play a role in sustaining catalepsy with both types of antipsychotics following chronic treatment. Supporting this notion, in the dorsal striatum, which is associated with EPS, the Olz group showed no such correlation. Interestingly, consistent with previous findings on acute treatment [25], the Clz group, which uniquely exhibited substantial EPS reduction after withdrawal, exhibited exclusive FosB-ir in the NAc, encompassing both the core and shell regions.

Robertson and co-workers [25] proposed that atypical antipsychotics lead to increased Fos-ir in the NAc Shell region, while an increase in Fos-positive neurons in the DL striatum is associated with EPS and is observed with typical antipsychotics [25,26]. This observation is supported by Sonego et al. [73], who associated EPS induced by Hal with the DL striatum [73]. While the DL striatum is typically implicated in motor control [74], activation of the DM striatum has also been described as a target for motor effects induced by atypical antipsychotic drugs [75].

In the present study, Hal, a well-known typical antipsychotic, was used as a proof of concept. Positive correlations were found between FosB-ir and Hal treatment, particularly in the DL and DM striatum regions, which were linked to movement-related symptoms like catalepsy and immobility. Negative correlations were observed between FosB-ir and total lines crossed in both Olz and Hal-treated groups, suggesting a shared pattern of activation, in movement-oriented tasks. These correlations remained consistent even after Hal withdrawal. Accordingly, prior research indicated FosB/ΔFosB staining in D2-MSNs after chronic Hal, tied to reduced movement motivation such as catalepsy [28,57,76,77]. Also, no specific changes in FosB expression were observed after chronic treatment with several protocols of withdrawal [78].

Nevertheless, marking predictions regarding FosB-ir changes due to chronic atypical antipsychotic treatment might be more intricate. This is particularly the case for FosB-ir in the NAc-Shell region, which seems to positively predict immobilization time or its reduction with Clz after the washout period, while not exhibiting a similar trend with Olz treatment. On the other hand, the Olz stable-cataleptic effect is correlated with NAc-Shell region, although dorsal striatum activity also should be considered, as confirmed by Hal. In fact, the immobility effect seems to be linked to NAc Shell region, as a locus of atypical antipsychotic action and mild-stress condition [79] evoked by the exposition to open field, a test used to evaluate locomotor activity and emotional patterns [80]. Indeed, NAc shell region is considered a limbic locus of antipsychotic action [25].

Interestingly, in the NAc, both atypical treatments led to an elevation in phosphoThr^34^–DARPP-32-ir, suggesting a potential role in modulating DA signaling in this region, an outcome not observed with Hal. Thus, the associated increase in FosB and phosphoThr^34^–DARPP-32 positive cells, in the NAc-Shell region, confirms the effect of both atypical antipsychotics in this region [61,81,82,83]. Once again, the analysis of results has revealed divergent patterns within this region, specifically concerning the induction of phosphoThr^34^–DARPP-32. In both atypical antipsychotics, the increase in phosphoThr^34^–DARPP-32 in the NAc Shell region can predict the effect on immobilization time, and the effect on lines crossing by Olz (negative correlation). In contrast, its reduced immunoreactivity following chronic Hal treatment correlates with behaviors that showed reduced activity (lines crossing and rota-rod test), alongside an escalated cataleptic effect and immobilization.

It is well-known that PKA increases phosphorylation of the phosphoThr^34^–DARPP-32 [31], and controls Fos-like expression in striatal projection pathways [29]. Possibly, the major finding of the present study was that only Clz treatment resulted in a response of phosphoThr^34^–DARPP-32 and FosB labeling in the NAc Core region, an area associated with sensorimotor integrative function [84], although significant correlation phosphoThr^34^–DARPP-32 was no observed. Interestingly, FosB-ir was not evident with this same drug, in the dorsal striatum, another sensorimotor region involved in the integration of the dopaminergic and glutamatergic neurotransmission originating mainly from cortical areas and correlated with the susceptibility of antipsychotic drugs to induce EPS [25]. As mentioned, Clz treatment was the unique treatment that resulted in a reduced effect on catalepsy response, following a short-withdrawal period, suggesting again that FosB-ir in dorsal striatum is implicated in EPS, and that possible activation of NAc Core region can prevent this effect, mainly for catalepsy and immobilization.

Increased FosB expression [29], and phosphoThr^34^–DARPP-32 [31], associated with catalepsy [36], after chronic administration, was only observed for Olz in the dorsal striatum and NAc Shell-region (phosphoThr^34^–DARPP-32 devoid of correlation), but this sequential activation was not observed with Hal treatment (phosphoThr^34^–DARPP-32, negative correlation for catalepsy and immobilization), suggesting that the unchanging cataleptic effect observed with Olz and Hal non necessarily shares a phosphoThr^34^–DARPP-32 immunoreactivity. However, an increased FosB-ir in the dorsal striatum (mainly DL) and NAc Shell region should be important. In addition, the similar trend of FosB-ir elicited by Olz administration shows that it shares some neural activation patterns and clinical features with Hal [20,85].

The involvement of DARPP-32 in the development of schizophrenia has been suggested [86], because this protein is deemed to be a representative marker for striatal projection neurons [87]. As mentioned, phosphorylation of DARPP-32 substrate at Thr^34^ residue by binding to D1R leads to subsequent activation of PKA, but, when phosphorylated at Thr^75^, by cyclin-dependent kinase 5 (CDK5), it is converted into an inhibitor of PKA [86]. In our experimental conditions, changes in phosphoThr^75^–DARPP-32 were solely observed with Clz treatment in the dorsal striatum. The reduction in catalepsy behavior, and absence of FosB staining, with decreased phosphoThr^75^–DARPP-32-ir was possibly observed because of the pharmacological mechanism of the Clz with short-term D2R occupancy, with a net rate of reversal of receptor blockade, widely known as the “fast-off D2” antagonism theory [65,71]. Previously, an effect on phosphoThr^75^ induced by Clz was described, potentially stemming from its influence on 5-HT_6_ receptors in both D1R and D2R neurons [36].

Here we suggest a hypothetical mechanism, where the short occupancy on D2R associated with a short Clz withdrawal period resulted in countered inhibitory pathway or more likely activation of NAc-Core region providing inhibition on pars compacta substantia nigra (SNc), thus, inactivating the nigrostriatal projection [88] (see Figure 6), with loss of modulation on the FosB dorsal striatum and neurochemical reduction of phosphoThr^75^–DARPP-32-ir. Besides, a previous study suggests that chronic, but not acute, repeated Clz administration with the same dosage used in the present study, may hyperactivate the calcineurin pathway [89], stimulating a type of calcineurin (PP2A, protein phosphatase-2A) and decreasing Thr^75^ phosphorylation [90], with the likelihood of calcineurin mediating the therapeutic effects of antipsychotics [89].

The differences found in the literature can be attributed to variations in mice strains, time courses, doses, route of administration, and withdrawal periods. This research exclusively used male mice, which stands as a limitation when extrapolating the findings to both sexes in humans, as gender differences are observed clinically. Nonetheless, a study involving chronic treatment holds the advantage of more accurately reflecting the therapeutic demands of these drugs, given that patients with schizophrenia require sustained daily medication. Evidence suggests that while therapeutic benefits may manifest early, the optimal clinical advantages of antipsychotics manifest with chronic usage [91], thereby aligning closely with our findings of EPS association after repeated treatment. Additionally, antipsychotics are multi-target drugs implying that changes in the phospho-DARPP32 protein might also encompass other receptors.

**Figure 6 biomedicines-11-02677-f006:**
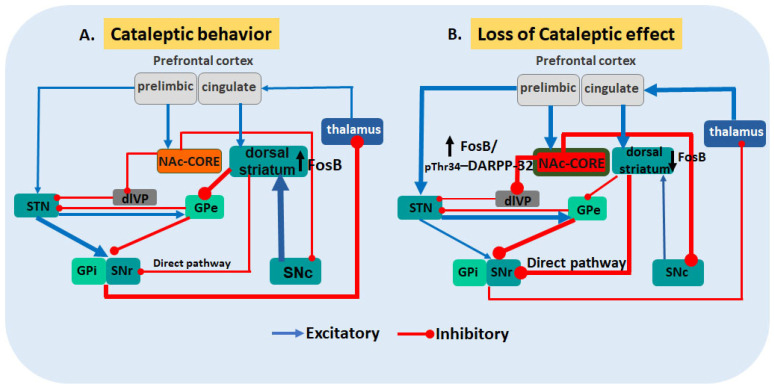
Hypothetical NAc-core region indirect output pathways as an outcome of the shift in the NAc FosB/phosphoThr^34^-DARPP-32-ir, reducing labeling FosB in DL-striatum, with Clz treatment. EPS induced by Clz was similar to Olz and Hal before the short-term withdrawal. After withdrawal, Clz was the only neuroleptic that reduced all behaviors including catalepsy. (**A**) Cataleptic behavior by Clz: showing NAc Core region as critical, where motivations derived from limbic regions interface with motor control circuitry to regulate appropriate goal-direct behavior [92]. Diagram illustrating activated NAc Core from previous prelimbic prefrontal cortex activation converge on the NAc Core inhibitory D2-MSNs output pathway to dorsolateral ventral pallidum nucleus (dlVP) whereby projects to subthalamic nucleus (STN) and sequential activation on the reticular part of the substantia nigra (SNr) and internal globus pallidum (GPi), mediating the inhibition of the thalamus that ultimately results in cataleptic behavior. (**B**) NAc Core is overstimulated, as observed by higher FosB-ir/phospho Thr^34^DARPP-32-ir. This results in increased inhibition over SNc [93,94], and subsequently weakened excitatory nigrostriatal projection, as observed by lack of FosB-ir in dorsal striatum after Clz, and labeling for phospho Thr^75^DARPP-32-ir, possibly with a tendency to counteracting the presence of catalepsy and immobilization, but probably increasing the direct pathway, as observed in akathisia disorder with Clz [37]. GPe: external globus pallidus.

## 5. Conclusions

This study provides the following pieces of evidence: (i) all three antipsychotics used in this study led to EPS following prolonged administration, but a washout period was critical for preventing Clz-induced side effects; (ii) after a washout period, the stable-cataleptic effect observed with Olz and Hal does not share a phosphoThr^34^–DARPP-32-ir, however, an increased FosB-ir in DL and NAc Shell-region seems to be important for immobilization; (iii) FosB/phosphoThr^34^-Darpp-32-ir in the NAc Core region is associated with hypokinetic movements inhibition and can be used to predict catalepsy and immobilization prevention, as observed with Clz. Consequently, considerations regarding the benefit/risk to EPS of these medications should be updated accordingly in psychiatry practice guidelines.

## Figures and Tables

**Figure 1 biomedicines-11-02677-f001:**
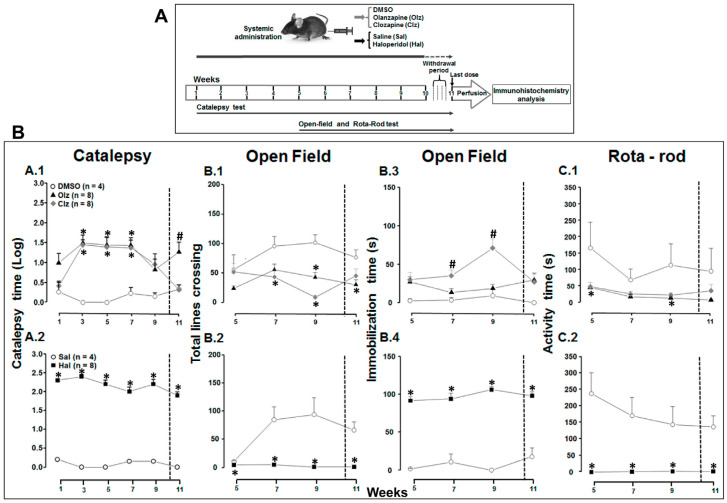
(**A**) Experimental design of the study. Male C57Bl/6J mice received daily i.p. injections (10 mL/kg) one day for ~10 weeks (72 days) of Olanzapine (Olz, 15 mg/kg), Clozapine (Clz, 20 mg/kg), or 25% DMSO solution (DMSO), and Haloperidol (Hal, 1 mg/kg) or Saline (Sal). A 4-day withdrawal period (days 73–76) followed chronic treatment. On day 77, all groups received one more treatment, and twenty-four hours later (day 78) the animals were deeply anesthetized and intracardially perfused for tissue collection for immunohistochemistry analyses. Catalepsy test was recorded from the first week of treatment, open-field and rota-rod test from the fifth week of treatment onwards. (**B**) Catalepsy and motor scores during long-term treatment up to 11 weeks. The measurement was 90 min after drug or vehicle injection. (**A.1**,**A.2**) Catalepsy time on 1, 3, 5, 7, 9, and 11 weeks of treatment, although the measurements were weekly. The results are expressed in Log(10). (**B.1**–**B.4**) Effects of antipsychotics on open field exploration during 5 min period from weeks 5 to 11 of treatment. (**B.1**,**B.2**) Total lines crossed indicating the number of floor areas entered with all four paws. (**B.3**,**B.4**) Immobility time indicating the time that the mouse remains without performing grooming, rearing, or exploring actions. (**C.1**,**C.2**) Effect of antipsychotics on Rota-rod test from weeks 5 to 11 of treatment. * significant difference when compared with the control group, respectively. # significant difference when compared with all groups. Two-way repeated-measures ANOVA followed by Bonferroni test (*p* < 0.05). Data are presented as the mean ± SEM. The dashed line indicates the withdrawal period.

**Figure 2 biomedicines-11-02677-f002:**
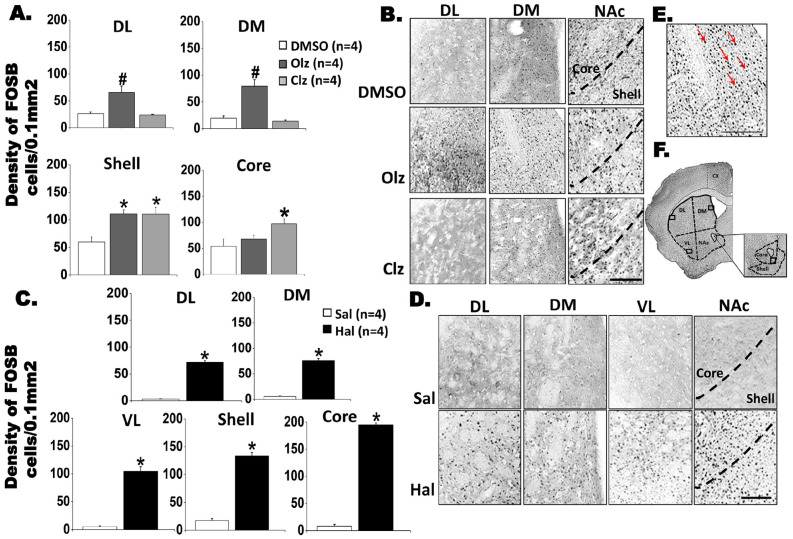
FosB-positive neurons in the striatum and NAc induced by chronic treatment with antipsychotics after 11 weeks of treatment (see Figure 1 for experimental design details). (**A**). Density FosB-positive cells and (**B**). Representative photomicrographs showing FosB-ir cells, after chronic treatment with Olz 15 mg/Kg or Clz mg/kg 20 mg/kg. * significant difference when compared with the control group. # significant difference when compared with all groups. One-way ANOVA followed by Bonferroni test (*p* < 0.05). (**C**). Density FosB-positive cells and (**D**). Representative photomicrographs showing FosB-ir cells after chronic treatment with Hal 1 mg/kg. (**E**). Insert with arrows showing an example of immunostained nuclei (grayscale) that can be distinguished. * Significant differences when compared with the control group, *t*-test, *p* < 0.05. Quantitative data are presented as mean ± SEM. (**F**). Insert shows the target area and highlights the target regions for photomicrographs (scale bar 50 µm). Olz: Olanzapine, Clz: Clozapine, Hal: Haloperidol, Sal: Saline. DL: dorsolateral striatum; DM: dorsomedial striatum; VL: ventrolateral striatum, and NAc: Nucleus accumbens.

**Figure 3 biomedicines-11-02677-f003:**
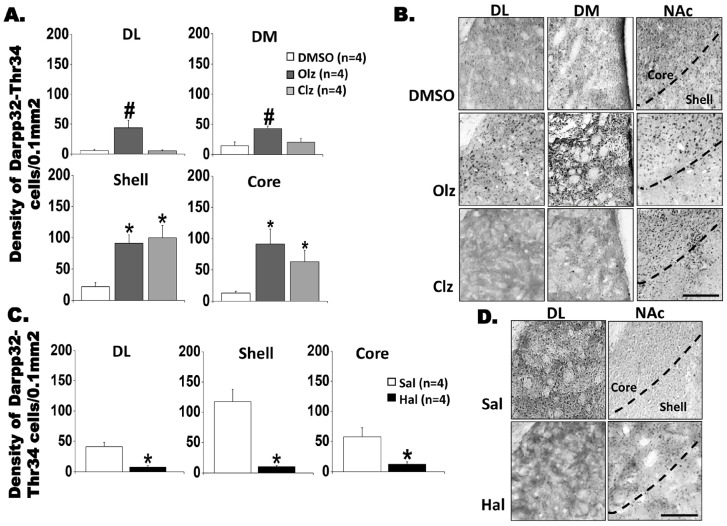
Thr^34^-DARPP-32^+^ cells in the striatum and NAc induced by chronic treatment with antipsychotics after 11 weeks (further specifications as in Figure 1 and Figure 2). (**A**). Graphs of density DARPP-32 phospho Thr^34^ cells and (**B**). Representative photomicrographs showing DARPP-32 phospho Thr^34^ positive cells, after chronic treatment with Olz 15 mg/Kg or Clz mg/kg 20 mg/kg. * significant difference when compared with the control group. # significant difference when compared with all groups. One-way ANOVA followed by Bonferroni test (*p* < 0.05). (**C**). Graphs of density DARPP-32 phospho Thr^34^ cells and (**D**). Representative photomicrographs showing DARPP-32 phospho Thr^34^-positive cells, after chronic treatment with Hal 1 mg/kg. Scale bar 50 µm. * significant differences when compared with the control group, *t*-test, *p* < 0.05. Quantitative data are presented as mean ± SEM. Olz: Olanzapine, Clz: Clozapine, Hal: Haloperidol, Sal: Saline. DL: dorsolateral striatum; DM: dorsomedial striatum and NAc: Nucleus accumbens.

**Figure 4 biomedicines-11-02677-f004:**
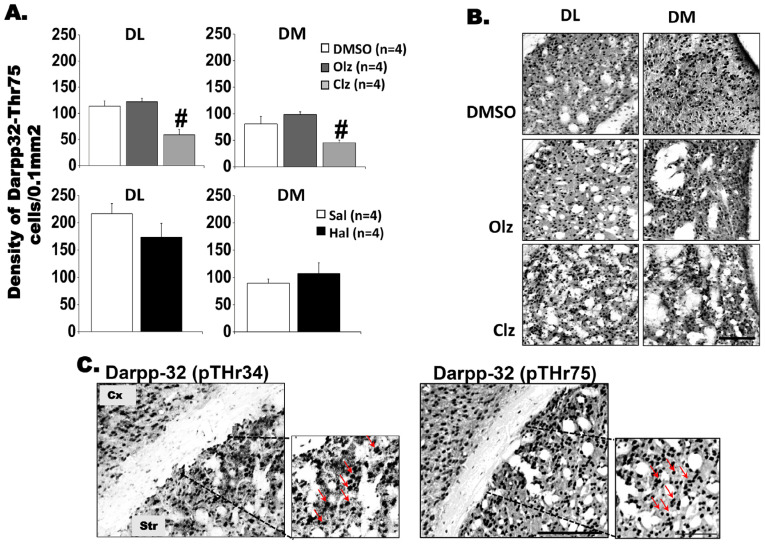
Thr^75^-DARPP-32^+^ cells in striatum induced by chronic treatment with antipsychotics after 11 weeks. (**A**). Graphs of the density of DARPP-32 phospho Thr^75^ cells and (**B**). Representative photomicrographs showing DARPP-32 phospho Thr^75^-positive cells, after chronic treatment with Olz 15 mg/Kg or Clz mg/kg 20 mg/kg in DL, DM regions. # significant difference when compared with all groups. One-way ANOVA followed by Bonferroni test (*p* < 0.05). Hal experiment, *t*-test, *p* > 0.05. (**C**). The photomicrographs display specific labeling of DARPP-32 phospho Thr^34^ and phospho Thr^75^ in the cortex and striatum. Inserts with arrows showing specific labeling. Scale bars 50 (**B**) and 100 µm (**C**). Quantitative data are presented as mean ± SEM. Olz: Olanzapine, Clz: Clozapine, Hal: Haloperidol, Sal: Saline. DL: dorsolateral striatum; DM: dorsomedial striatum; Cx: Cortex and Str: Striatum.

**Figure 5 biomedicines-11-02677-f005:**
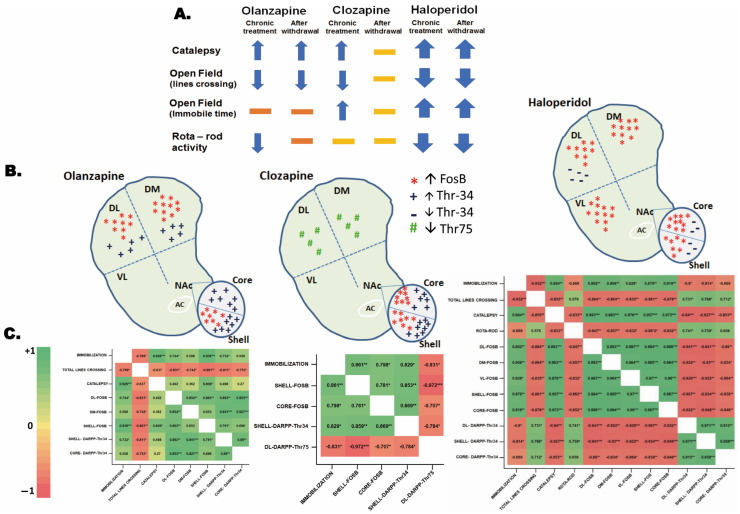
(**A**) Schemes with an overview of behavior, and (**B**) significant alterations protein-ir in comparison to control respectively. (**C**) Heatmap of immunoreactivity for FosB and phospho Thr^34^ or Thr^75^-DARPP-32, with correlation coefficient (r) for significant subsets of each behavior and each protein-ir (Pearson’s r matrix). The color range shows green indicating a higher positive correlation and red indicating a higher negative correlation. * *p* < 0.05; ** *p* < 0.01; *** *p* < 0.001.

## Data Availability

Data supporting reported results can be found in https://drive.google.com/drive/folders/1cttMsvT2Xc8IZUC1Xd34-jiqIAKevEjj?usp=drive_link (accessed on 25 August 2023).

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
