# Peer review of "Extrapyramidal Side Effects with Chronic Atypical Antipsychotic Can Be Predicted by Labeling Pattern of FosB and phosphoThr34-DARPP-32 in Nucleus Accumbens"

_biomedicines, 2023, doi:10.3390/biomedicines11102677_

Round 1

Reviewer 1 Report

The present study describes the various effects produced by the antipsychotics in mice, but it does not fully explain the implications of these findings or how they contribute to the understanding of EPS in humans.

Specific comments:

1. The abstract mentions that motor activity was determined throughout the treatment, but it does not specify how this was measured. The lack of details makes it difficult to assess the reliability and relevance of the motor activity results.

2. "Nevertheless FosB/phosphoThr34-Darpp-32-ir in NAc Core-region can is ..." - so is this 'can' or 'is'?

3. In the introduction, it is relevant to mention that when faced with a patient that does not respond to or cannot tolerate standard antipsychotic therapy, physicians have limited treatment strategies (citation: pubmed.ncbi.nlm.nih.gov/32728915). It does not help that a significant proportion of patients with schizophrenia do not respond sufficiently to antipsychotics and combination therapy is often required. For patients with inadequate response in target psychotic or schizophrenia symptoms, adding a second antipsychotic medication has yielded largely negative results based on randomized trials (citation: pubmed.ncbi.nlm.nih.gov/15625211).

4. It is important to at least briefly explain the rationale between choosing Olanzapine, Clozapine, and Haloperidol for study.

5. Given the difference in metabolism between rats and humans, how were the drug dosages extrapolated and determined in this study?

6. "2.5. Quantification" - quantification of? Please be specific.

7. "The Pearson correlation coefficient (Pearson's r, range of values from +1 to -1) ..." - please standardize the font choice and font size throughout the manuscript.

8. Antipsychotic-related extrapyramidal symptoms result from the disruption of dopamine neurotransmission in the brain, particularly the blockade of D2 dopamine receptors. These symptoms can vary in type and severity and may occur shortly after starting treatment or, in the case of tardive dyskinesia, after prolonged use of antipsychotic medications. It is important to make the distinction between the type of extrapyramidal symptoms studied and shown here.

9. What does 'CORE' in Figure 6 refer to? Further clarification is necessary.

10. The study limitations should be discussed. For example, the study was conducted on male C57Bl/6J mice. Extrapolating the findings to humans might be a limitation due to species and gender differences in responses to medications. 

Moderate edits required.

Author Response

Referee # 1

We would like to extend our sincere gratitude to the reviewer for the invaluable contributions and constructive comments on our scientific manuscript. Your expertise and thoughtful feedback have played a pivotal role in improving the quality and rigor of our research. We want to clarify that references cited in the text were thoroughly reviewed or updated. Please, find below a point-by-point response to the specific comments raised.

  1. The abstract mentions that motor activity was determined throughout the treatment, but it does not specify how this was measured. The lack of details makes it difficult to assess the reliability and relevance of the motor activity results.

Answer: According to the referee suggestion, more details were added to clarify this issue.

“… for a period of 11-weeks with a 4-day withdrawal period before the last dosage. Catalepsy for detection of EPS, along with open-field, and rotarod tests, were assessed as behavioral correlates of motor responses. Additionally, FosB and phosphorylated -DARPP-32 immunohistochemistry were examined in striatal regions after treatment. All antipsychotics produced catalepsy, reduced open-field exploration, as such as impaired rota-rod performance after Olz and Hal.”

(Please, see Pages 1-2)

  1. "Nevertheless FosB/phosphoThr34-Darpp-32-ir in NAc Core-region can is ..." - so is this 'can' or 'is'?

Answer: Thank you for sharing the sentence. The correct form should be 'is' instead of 'can.' We apologize for the confusion. So, the revised sentence reads: 'Nevertheless, FosB/phosphoThr34-Darpp-32-ir in NAc Core-region is…’

(Please see Page 2)

  1. In the introduction, it is relevant to mention that when faced with a patient that does not respond to or cannot tolerate standard antipsychotic therapy, physicians have limited treatment strategies (citation: pubmed.ncbi.nlm.nih.gov/32728915). It does not help that a significant proportion of patients with schizophrenia do not respond sufficiently to antipsychotics and combination therapy is often required. For patients with inadequate response in target psychotic or schizophrenia symptoms, adding a second antipsychotic medication has yielded largely negative results based on randomized trials (citation: pubmed.ncbi.nlm.nih.gov/15625211).

Answer: Your comments have been instrumental in enhancing the quality and relevance of our work. We acknowledge the importance of the points raised, and we have made the following modifications to address these concerns. In the introduction, we have incorporated your valuable suggestion to mention the limited treatment strategies for patients who do not respond to or cannot tolerate standard antipsychotic therapy (citation: pubmed.ncbi.nlm.nih.gov/32728915). Additionally, we have highlighted the challenge of inadequate response to antipsychotic treatment and the largely negative results of adding a second antipsychotic medication based on randomized trials (citation: pubmed.ncbi.nlm.nih.gov/15625211). We believe these additions strengthen the context and significance of our study. Once again, we thank the reviewer for their constructive input.

  • EPS are a leading cause of noncompliance with antipsychotic medication which is associated with a relapse of psychotic symptoms and re-hospitalization [5]. In clinical practice, treating schizophrenia remains a significant challenge, especially in patients who do not respond to or cannot tolerate standard antipsychotic therapy [6]. Physicians have limited treatment strategies in such case [6].

(Please, see Page 2)

  • … EPS must be weighed against the acknowledged potential for metabolic syndrome [37–39]. Besides, in medical practice, a considerable number of individuals with schizophrenia do not respond adequately to antipsychotic medication alone, often requiring a combination of second-generation antipsychotics [40,41]. In this situation, addition of a second antipsychotic has shown different outcomes needing more rigorous randomized trials [40].

(Please, see Page 4)

  • Then, the importance of separately studying downstream effector proteins in MSNs becomes evident due to variations in the occurrence of EPS among these types of antipsychotics, which may play a role in unfavorable synergistic actions during a dual therapy. These three medications shown different affinity and dissociation from D2R, and by comparing their effects, we aim to gain insights into potential variations in treatment response and side effects among individuals. This information is critical for optimizing treatment strategies. In fact, comparing the differences in measurements of extrapyramidal symptoms between Olz versus Clz, and Hal is associated with safety liabilities.

(Please, see Page 4)

  1. It is important to at least briefly explain the rationale between choosing Olanzapine, Clozapine, and Haloperidol for study.

Answer: Thank you for highlighting the need for clarification on this point. We have added a paragraph with the rationale for selecting Olanzapine, Clozapine, and Haloperidol.

  • Then, the importance of separately studying downstream effector proteins in MSNs becomes evident due to variations in the occurrence of EPS among these types of antipsychotics, which may play a role in unfavorable synergistic actions during a dual therapy. These three medications shown different affinity and dissociation from D2R, and by comparing their effects, we aim to gain insights into potential variations in treatment response and side effects among individuals. This information is critical for optimizing treatment strategies. In fact, comparing the differences in measurements of extrapyramidal symptoms between Olz versus Clz, and Hal is associated with safety liabilities.

(Please, see Page 4).

  1. Given the difference in metabolism between rats and humans, how were the drug dosages extrapolated and determined in this study?

Answer: This is a very important point raised by the reviewer. As mentioned in the manuscript, the objective was to study EPS-induced by antipsychotics, and an initial objective with tardive dyskinesia (TD). This is a pre-clinical study. Then, the selected doses were based on animal’s studies, with citations provided for each drug. For Olz , Turrone et al., 2005 (DOI: 10.1016/j.biopsych.2004.10.023), and for Clz and Hal, Sebens et al., 1998 (DOI: 10.1016/s0014-2999(98)00391-4) are well-recognized in pre-clinical studies.

Other references with the same doses were cited throughout the text.

However, here, it is being described how to convert the dose used in a mouse to a dose in mg/kg for humans (https://www.ncbi.nlm.nih.gov/pmc/articles/PMC4804402/pdf/JBCP-7-27.pdf, table 1): divide animal dose by 12.3 or multiply animal dose by 0.081 (Values based on data from FDA Draft Guidelines).

  • Olanzapine (Zyprexa) = (15 mg/kg)/12.3 = 1.2 mg/kg (therapeutic dose in human is approximately 0.33 mg/kg, see pharmacokinetics/toxicology: https://www.ema.europa.eu/en/documents/scientific-discussion/zyprexa-epar-scientific-discussion_en.pdf). The dose used in our study had already been explored in toxicokinetic studies with animals.
  • Clozapine (Leponex) = (20 mg/kg)/12.3 = 1.62 mg/kg (see table 1, intramuscular administration in https://www.ncbi.nlm.nih.gov/pmc/articles/PMC8464180/pdf/cureus-0013-00000018267.pdf)
  • Haloperidol (Janssen Drops) = (1 mg/kg)/12.3 = 0.081 mg/kg (haloperidol 5-10 mg IM or IV, in https://pubmed.ncbi.nlm.nih.gov/32340820/)

Thus, in section 2.2. Drug administration we included the sentence and reference: “The doses used for Clz and Hal fall within a human dosage range (see calculation in Nair and Jacob, 2016). For Olz, the dose was based in toxicokinetic studies in rats.”

(Please, see Page 5)

  1. "2.5. Quantification" - quantification of? Please be specific:

Answer: 2.5. Quantification of immunohistochemistry

(Please, see Page 8)

  1. "The Pearson correlation coefficient (Pearson's r, range of values from +1 to -1) ..." - please standardize the font choice and font size throughout the manuscript:

Answer: Sorry for the inconsistencies in font size. We have carefully checked to ensure its consistency.

  1. Antipsychotic-related extrapyramidal symptoms result from the disruption of dopamine neurotransmission in the brain, particularly the blockade of D2 dopamine receptors. These symptoms can vary in type and severity and may occur shortly after starting treatment or, in the case of tardive dyskinesia, after prolonged use of antipsychotic medications. It is important to make the distinction between the type of extrapyramidal symptoms studied and shown here.

Answer: We have edited discussion initial paragraph, according to the suggestion.

It is reported that typical and atypical antipsychotics differ in their ability to induce EPS. Antipsychotic-related extrapyramidal symptoms result from the disruption of dopamine neurotransmission in the brain, particularly the blockade of D2 dopamine receptors. These symptoms can vary in type and severity and may occur shortly after starting treatment or, in the case of tardive dyskinesia, after prolonged use of antipsychotic medications.

(Please, see Page 19)

  1. What does 'CORE' in Figure 6 refer to? Further clarification is necessary.

Answer: CORE in Figure 6, means core region of NAc. In this version of the manuscript, we assured it was correctly stated in the manuscript, as well as specified in the legend and in the Figure 6. Also, inside the figure was placed increase of FosB/pThr34-DARPP32

Figure 6. Hypothetical NAc-Core region indirect output pathways as outcome of the shift in the NAc FosB/phosphoThr34-DARPP-32-ir, reducing labeling FosB in DL-striatum, with Clz treatment.

(Please, see Page 25)

  1. The study limitations should be discussed. For example, the study was conducted on male C57Bl/6J mice. Extrapolating the findings to humans might be a limitation due to species and gender differences in responses to medications.

Answer: According to the suggestion, we have added, in the last paragraph before Conclusion, a statement mentioning limitations of the study.

The differences found in the literature can be attributed to variations in mice strains, time courses, doses, route of administration, and withdrawal periods. This research exclusively used male mice, which stands as a limitation when extrapolating the findings to both sexes in humans, as gender differences are observed clinically.

(Please, see Page 24)

Reviewer 2 Report

In this study, the authors tried to determine what differential and long-lasting changes in FosB and DARPP-32 phosphorylation immunoreactivity (-ir) are present in the striatum and Nucleus accumbens (NAc) following hypokinetic movement disorders associated with antipsychotic-induced extrapyramidal side effects (EPS). The authors suggested that atypical antipsychotics Olz and Clz also give rise to EPS effects frequently associated with a cumulative dosage of typical neuroleptics such as Hal, and FosB/phosphoThr34-Darpp-32-ir in NAc Core-region can is associated with hypokinetic movements inhibition.

Comments

The reviewer has some concerns as follows:

1. Please provide the information for sources of animals and tested drugs.

2. Please check that the font sizes need to be consistent, such as lines 242, 247-249, 412, 477-479 and others.

3. The presentations for Figures 2-4 are not convincing. It is difficult to identify where is the positive staining from the black and white photos. Moreover, how to quantification for these IHC results?

4. The correlations are not convincing. Can an association be inferred from only the behavioral and immunohistochemistry results? The evidence seems a bit weak.

Author Response

Referee # 2

We wish to express our deep appreciation to the reviewer for the invaluable contributions and insightful comments provided on our manuscript. Your dedication has significantly strengthened the quality and impact of our research, and we are sincerely grateful for the time and expertise you invested in reviewing our work. We would like to emphasize that we have thoroughly reviewed and updated all the references cited in the text. Below, you will find a comprehensive, point-by-point response addressing the specific comments raised."

  1. Please provide the information for sources of animals and tested drugs.

Answer:  The information is now provided in section 2.1 of the methods:

The thirty-two-adult male C57Bl/6J mice, 8- weeks old at the beginning of the experiments, weighing 25-30g. Animals were obtained from Jackson Lab's colonies housed at UFABC Animal House in a temperature-controlled room (23°C) with a 12-hour light/dark cycle (lights on at 7:00 a.m.) and with unlimited access to water and food.

(Please, see Page 8).

  • Please, in section 2.2 the information about the drugs is already cited.

Olanzapine (Olz, 15 mg/Kg, Zyprexa tablet®

Clozapine (Clz, 20 mg/Kg, Leponex tablet®

Haloperidol (Hal, 1 mg/Kg, Janssen Drops®

(Please, see Page 8).

  1. Please check that the font sizes need to be consistent, such as lines 242, 247-249, 412, 477-479 and others.

Answer: We apologize for the lack of consistency in font sizes. We have checked it carefully and made it consistent throughout the text.

  1. The presentations for Figures 2-4 are not convincing. It is difficult to identify where is the positive staining from the black and white photos. Moreover, how to quantification for these IHC results?

Answer: We appreciate your feedback regarding the presentations of Figures 2-4. To address this concern, we have enhanced the sharpness in the images, and we have also added arrows to clearly indicate the areas of positive staining. We believe that these adjustments will significantly improve the clarity and interpretability of the figures, making it easier to identify the positive staining within the black and white photos. Thank you for bringing this to our attention, and we hope that the revised figures meet your expectations.

In regard to the quantification method used, we have improved the separate section in the manuscript detailing the quantification methods used for the IHC results, ensuring transparency and accuracy in our analysis. We believe that these enhancements will better convey our findings and address your concerns. Thank you for your valuable input.

“A preliminary qualitative analysis of all brain sections was conducted to identify bilateral labeling and brain structures. Structure localization was determined with the help of the Paxinos and Franklin atlas [55]. Images containing immunostained nuclei or cells from slices were digitally captured with 10 X magnification through a Leica DFC295 camera connected to the 5500 DM B Leica® microscope. A computerized image analysis system (ImageJ, NIH System – https://imagej.nih.gov/ij/download.html) was used. The striatum was subdivided into three regions for analysis: Dorsolateral (DL), Dorsomedial (DM) and Ventrolateral (VL) (Fig. 2-F) [56]. The NAc was subdivided into Core and Shell regions (Fig. 2-F) [57]. For FosB/ΔfosB immunoreactivity (ir) labeling was visualized by a higher brown reaction product, and cytoplasmic DARPP-32 positive cells were visualized as a dark blue reaction product [34]. At least three bilateral sections from each animal were selected for counting the number in each region of positive nuclei or cells, according to corresponding coordinates on the atlas along an anterior-posterior (AP) axis (bregma 0.86, 0.74 and 0.62 mm). The counting was automated using ImageJ following the basic image processing from the software. To ensure the validity of the results and minimize bias, two different experimenters conducted the counting independently while blinded to the data. The results were subsequently cross-validated for accuracy. Results are expressed as the mean density of cells (number of positive nucleus or cells/0.1 mm2 of the region), calculated from results obtained in each brain side (Dos-Santos-Pereira et al., 2016). Four animals per group were included based on established criteria for IHS and quantification. The quantitative analysis was conducted blindly.”

(Please, see Page 11)

  1. The correlations are not convincing. Can an association be inferred from only the behavioral and immunohistochemistry results? The evidence seems a bit weak.

Answer: We appreciate your feedback and concern regarding the correlations and the strength of evidence in our study. We understand the importance of critically evaluating research findings. Nevertheless, we would like to highlight the significance of utilizing the correlation between behavioral and immunohistochemistry data in our study.

The use of correlations between behavioral and immunohistochemistry results is a common approach in many scientific studies. Indeed, the correlation between immunohistochemistry and behavior is a valid approach as it bridges the gap between molecular and behavioral levels of analysis. By linking specific molecular markers to observed behaviors, it offers valuable insights into the underlying biological mechanisms contributing to behavioral outcomes, enhancing our understanding of complex physiological processes.

In our study, we were stringent about what behaviors to test for the correlations by only evaluating the cases where significant differences in the behavior were confirmed. To do this, we applied traditional and well-known statistics (t-test, ANOVAS, Post Hoc Bonferroni, effect size ω²) to confirm the results of behavior (RM-ANOVA, main effect, Post Hoc Bonferroni). After verifying the significant differences, linear regressions were performed to determine the relationship, whether positive or negative, between behaviors and labeling. The heatmap correlation matrix displays the positive or negative correlations between multiple variables as a color-coded matrix, also serving as a summary. Thus, we maintained the traditional and required statistical rigor.

Finally, we would like to emphasize that we are no longer drawing direct conclusion based solely on the correlation results. Therefore, we have revised our conclusion section accordingly:

“This study provides the following pieces of evidence: (i) all three antipsychotics used in this study led to EPS following prolonged administration, but a washout period was critical for preventing Clz-induced side effects; (ii) after a washout period, the stable-cataleptic effect observed with Olz and Hal does not share a phosphoThr34–DARPP-32-ir, however, an increased FosB-ir in DL and NAc Shell-region seems to be important for immobilization; (iii) FosB/phosphoThr34-Darpp-32-ir in NAc Core-region is associated with hypokinetic movements inhibition and can be used to predict catalepsy and immobilization prevention, as observed with Clz. Consequently, considerations regarding the benefit/risk to EPS of these medications should be updated accordingly in psychiatry practice guidelines.”

(Please, see Pages 28-29)

Round 2

Reviewer 1 Report

Thank you for the replies and revisions.

Some copyediting required.

Reviewer 2 Report

This revised manuscript has greatly improved. It can be accepted. No further comments.